# Multi-view Feature Extraction via Tunable Prompts is Enough for Image Manipulation Localization

Xuntao Liu
School of Computer
Science, Fudan University
Shanghai, China
xtliu22@m.fudan.edu.cn

Yuzhou Yang
School of Computer
Science, Fudan University
Shanghai, China
yangyz22@m.fudan.edu.cn

Haoyue Wang
School of Computer
Science, Fudan University
Shanghai, China
haoyuewang23@m.fudan.edu.cn

Qichao Ying
NVIDIA
Shanghai, China
shinydotcom@163.com

Zhenxing Qian[*]
School of Computer
Science, Fudan University
Shanghai, China
zxqian@fudan.edu.cn

Xinpeng Zhang
School of Computer
Science, Fudan University
Shanghai, China
zhangXinpeng@fudan.edu.cn

Sheng Li
School of Computer
Science, Fudan University
Shanghai, China
lisheng@fudan.edu.cn

## Abstract

Deceptive images can quickly spread via social networking services, posing significant risks. The rapid progress in Image Manipulation Localization (IML) seeks to address this issue. However, the scarcity of public training datasets in the IML task directly hampers the performance of models. To address the challenge, we propose a Prompt-IML framework, which leverages the rich prior knowledge of pre-trained models by employing tunable prompts. Specifically, sets of tunable prompts enable the frozen pre-trained model to extract multi-view features, including spatial and high-frequency features. This approach minimizes redundant architecture for feature extraction across different views, resulting in reduced training costs. In addition, we develop a plug-and-play Feature Alignment and Fusion module that seamlessly integrates into the pre-trained models without additional structural modifications. The proposed module reduces noise and uncertainty in features through interactive processing. The experimental results showcase that our proposed method attains superior performance across 6 test datasets, demonstrating exceptional robustness.

## CCS Concepts

• **Security and privacy** → **Intrusion/anomaly detection and malware mitigation**; • **Computing methodologies** → **Artificial intelligence**.

## Keywords

Image Manipulation Localization, Tunable Prompts, Attention Mechanism

[*]Corresponding author.

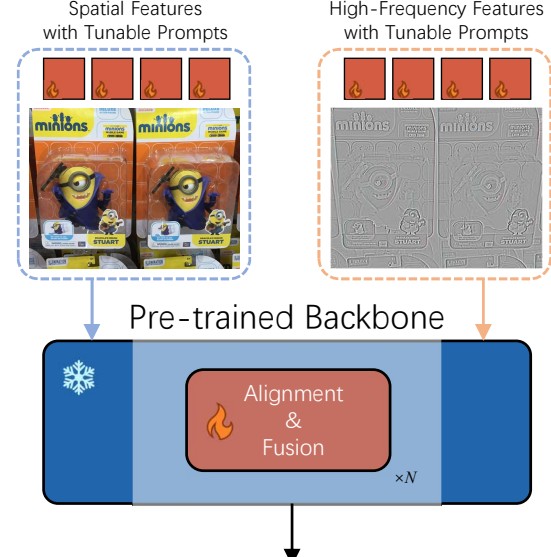

Image Manipulation Localization

**Figure 1: Prompt-IML utilizes a single pre-trained backbone with frozen parameters to handle multi-view features via tunable prompts. The Feature Alignment and Fusion module is designed as a plug-and-play component for feature interaction and enhancement.**

**ACM Reference Format:**
Xuntao Liu, Yuzhou Yang, Haoyue Wang, Qichao Ying, Zhenxing Qian, Xinpeng Zhang, and Sheng Li. 2024. Multi-view Feature Extraction via Tunable Prompts is Enough for Image Manipulation Localization. In *Proceedings of the 32nd ACM International Conference on Multimedia (MM '24), October 28-November 1, 2024, Melbourne, VIC, Australia.* ACM, New York, NY, USA, 9 pages. https://doi.org/10.1145/3664647.3681137

## 1 Introduction

With the evolution of image editing techniques, individuals can now freely edit images while preserving high quality. Commonly encountered methods such as copy-move, splicing, and inpainting have the potential to modify the original semantic content of images.

                                                    

The rapid progress of image editing tools significantly reduces the difficulty and cost of creating deceptive images. Therefore, deceptive images widespread on the internet, posing a significant social threat. In response, the Image Manipulation Localization (IML) task is widely employed to address these challenges.

Advancements in deep learning lead researchers to develop numerous manipulation localization networks [4, 8, 14, 19, 29, 31]. However, the performance of these models is limited by the scarcity of large-scale training datasets[1, 24, 37]. For example, the widely utilized CASIA2[5] dataset comprises only 7491 authentic images and 5063 forged images. To address this limitation, some researchers [4, 19, 29, 31] create extensive private training datasets by employing sophisticated data augmentation techniques on part of public datasets, e,g, COCO [17]. Other researchers attempt to weaken tampering traces and generate realistic manipulated images using methods such as adversarial network [37] and style transfer [1]. While these approaches improve model performance to some extent, the challenges of accessibility to many private datasets and the cost of manually creating manipulated images persist.

We observe that tasks such as classification, object detection, and semantic segmentation boast many pre-trained models endowed with rich prior knowledge, e.g. Swin-Transformer [20]. It is natural to consider leveraging these pre-trained models to address the challenges in IML task. However, directly applying them to IML task is proven inefficient [22]. This inefficiency stems from the unique nature of IML task, which focuses on extracting non-semantic visual cues and low-level discontinuities from images. Two key aspects illustrate this distinctiveness: 1) High-frequency information: images captured by different cameras exhibit varying noise patterns [16]. This brings inconsistent noise into forged images when manipulated and authentic areas come from different images. Moreover, images generated by different networks may manifest differences in the frequency domain [27]. 2) Edge information: the level of image editing can vary, leading to jagged and non-smooth edges at the boundary of forged area or color inconsistency [37]. These details are critical for precise manipulation localization but are frequently neglected in many tasks.

IML-ViT [22] is the pioneering attempt to employ pre-trained models based on the plain ViT[6] architecture in the IML task. They also incorporate edge supervision to direct the network's attention toward subtle forgery artifacts. However, IML-ViT overlooks the high-frequency information that has been validated effective in many previous works [4, 14, 15]. In IML task, processing multi-view features often requires parallel backbone architectures [4, 14], which becomes challenging as the emergency of models with increasing parameters. Additionally, IML-ViT, despite leveraging pre-trained models, necessitates training the model with datasets from the scratch. This undoubtedly places a substantial demand on computational resources, particularly for tuning large pre-trained models. Moreover, some previous works show that tuning large pre-trained models on downstream tasks may harm the performance of models [30], which is also observed during our comparison experiments.

In this paper, we propose Prompt-IML, as shown in Fig. 1, aiming to address the scarcity of datasets in IML task by leveraging the rich prior knowledge of pre-trained models. Specifically, Prompt-IML

follows an encoder-decoder architecture. An encoder based on a pre-trained model is responsible for feature extraction, and then a decoder processes these features to accurately locate manipulated regions. To process multi-view features beneficial for the IML task without resorting to complex parallel architectures, we propose employing sets of tunable prompts for exploiting the pre-trained model as the encoder. We freeze the pre-trained model while training these prompts. It offers three main advantages. Firstly, it allows the pre-trained model to be adapted for processing features from each view. Secondly, the processed features retain the robustness inherited from the pre-trained model. Lastly, it helps reduce the computational resources required for training.

Furthermore, considering the variations among multi-view features, we propose a Feature Alignment and Fusion (FAF) module. This module is designed as a plug-and-play component that can be seamlessly integrated into the encoder without additional structural modifications. Within the FAF module, multiple attention mechanisms are employed for different merits. The FAF module reduce noise and uncertainty in features, while also suppressing sporadic positive responses to ensure a consistent output.

To fairly evaluate the model's capabilities, we follow the evaluation protocol outlined in IML-ViT. It involves using only the CASIA2 dataset for training and then testing on the other 6 datasets. Importantly, we ensure zero data overlap between the training dataset and test datasets, making this a cross-dataset evaluation. Experimental results demonstrate that the proposed Prompt-IML effectively leverages the prior knowledge in pre-trained models, outperforming previous state-of-the-art methods and exhibiting stronger robustness. Our contributions can be summarized in three aspects:

- We introduce Prompt-IML to tackle the challenge posed by the scarcity of IML datasets. Our approach extracts and adjusts multi-view features from a single pre-trained backbone through the integration of tunable prompts, thereby preserving performance and robustness.
- We carefully craft a plug-and-play Feature Alignment and Fusion (FAF) module that seamlessly integrates into the backbone. It efficiently reduces noise and uncertainty in features while mitigating the impact of sporadic positive responses.
- Prompt-IML outperforms state-of-the-art methods across 6 test datasets. Our extensive experiments confirm the generalizability and robustness of our approach, and also validate the effectiveness of the proposed FAF module.

## 2 Related Works

### 2.1 Image Manipulation Localization

With the advancements in deep learning, researchers delve into developing end-to-end manipulation localization networks[2, 4, 10, 11, 14, 15, 19, 33–35, 37]. MVSS-Net++[4] integrates multi-scale features, edge-related features, and high-frequency features of images for feature extraction and utilizes spatial-channel attention for feature fusion and enhancement. PSCC-Net[19] proposes a progressive spatial-channel attention module, utilizing multi-scale features and dense cross-connections to generate tampering masks of various granularity. These works all employ CNN architecture as the

backbone, primarily due to the convolutional layers' local receptive fields being more conducive to dense predictions[20].

However, in various other tasks like image classification, the ViT architecture emerges as dominant. Some researchers are actively exploring ViT's application in IML task due to its global attention mechanism, which is particularly effective for modeling relationships between any regions regardless of their visual semantic relevance. ObjectFormer[31] constructs ViT-related architecture, utilizing a set of learnable object prototypes as mid-level representations to capture object-level consistencies across different regions. IML-ViT[22] is the first to incorporate a pre-trained ViT model into IML task and fine-tune the model using only CASIA2 dataset. This not only addresses the issue of data scarcity in the IML task but also reduces the cost of training the model. This work demonstrates the tremendous potential of pre-trained models based on the ViT architecture for IML task, offering an alternative approach to addressing the challenge of data scarcity.

### 2.2 Fine-tuning Methods

The primary purpose of fine-tuning is further training pre-trained models with small-scale datasets, aiming at adapting models to downstream tasks. This method leverages rich prior knowledge in pre-trained models, leading to faster convergence during training. The common method is full-tuning, which involves adjusting all parameters in the pre-trained model. However, it faces challenges in adapting to the emergence of models with increasing parameters due to the more computational resources required.

Compared to full-tuning, prompt-tuning is an efficient, low-cost way of adapting pre-trained models to downstream tasks. This technique is first used in NLP, and VPT[12] is an efficient way to adapt it for the visual domain. It inserts a small number of tunable prompts into the pre-trained model's input and adjusts the original features through self-attention mechanisms. Recently, EVP[18] attempts to apply visual prompts to low-level structure segmentation tasks, including IML task. They achieve precise localization of manipulated regions by adjusting the embedding representation of images and incorporating high-frequency information. However, their method of combining multi-view features through a basic addition strategy is considered inefficient due to potential variations among different features. Our experimental section demonstrates the inefficiency of their approach in IML task.

## 3 Proposed Method

### 3.1 Approach Overview

Fig. 2 illustrates the pipeline design of the proposed Prompt-IML, which follows the common Encoder-Decoder framework. The complete pipeline consists of two phases, i.e., feature extraction and manipulation localization. In the feature extraction stage, we employ the pre-trained Swin-Transformer as the backbone and keep its parameters frozen during training. Simultaneously, we utilize multiple sets of tunable prompts to adjust the spatial and high-frequency features of the images respectively. This approach thereby avoids employing redundant model architectures to extract features from additional views. Considering differences of multi-view features, we propose a Feature Alignment and Fusion (FAF) module for processing. FAF modules are integrated between the layers of the backbone,

effectively reducing noise and uncertainty within extracted features of each layer. Meanwhile, they help suppress sporadic positive responses, leading to more consistent output. These modules are plug-and-play, requiring no modifications to the backbone itself. In the manipulation localization stage, we employ Mask2Former as the decoder, which includes both a Pixel Decoder and a Transformer Decoder. The decoder processes the multi-scale features acquired from previous stage and produces the final prediction.

### 3.2 Feature Extraction Stage

We denote the input image as $\mathbf{X} \in \mathbb{R}^{h \times w \times 3}$. To acquire the input for spatial features, we partition the image into specified-sized patches:

$$\mathbf{F}_0^{RGB} = \text{Norm}(\text{Conv}(\mathbf{X})) + \mathbf{F}_{PE}, \tag{1}$$

where $\mathbf{F}_0^{RGB} \in \mathbb{R}^{H \times W \times C}$, Conv represents the partition operation, $\mathbf{F}_{PE}$ is a learnable positional embedding. Next, we employ a set of BayarConv with varying size kernels to extract high-frequency features:

$$\mathbf{F}_0^{HFQ} = \text{Concat}(\{\text{BayarConv}_{i \times i}(\mathbf{X})\}), \ i \in \{3, 5, 7\}, \tag{2}$$

where $\mathbf{F}_0^{HFQ} \in \mathbb{R}^{H \times W \times C}$, and $i$ symbolizes the kernel size. The obtained features will be sent to the backbone for further processing.

*3.2.1 Multi-view Features Processing with Tunable Prompts.* We employ the pre-trained Swin-Transformer, which is commonly used in the semantic segmentation (SS) task, as the backbone for the following reasons: 1) The Swin-Transformer includes a window attention design that has linear time complexity compared to image size; 2) Patch merging operations can generate multi-scale feature maps, which is proven to be important in IML task[4, 11]. 3) SS task and IML task share some similarities, as they are fundamentally pixel-level classification tasks. We believe that pre-trained models used for SS task, after fine-tuning, are more advantageous in achieving precise pixel-level manipulation localization.

The Swin-Transformer consists of 4 layers and outputs features with specific resolutions. We denote the output features at the $i$-th layer as $\mathbf{F}_i$:

$$\mathbf{F}_i = \text{Layer}_i(\mathbf{F}_{i-1}) \in \mathbb{R}^{(H_i \times W_i) \times C_i}, i \in \{1, 2, 3, 4\}, \tag{3}$$

where $H_i = \frac{H}{2^{i-1}}, W_i = \frac{W}{2^{i-1}}, C_i = C * 2^{i-1}$, $\text{Layer}_i$ symbolize the $i$-th layer of Swin-Transformer.

We adopt a prompt-tuning method[12] to enable a single pre-trained model to process both spatial and high-frequency features. Specifically, during training, we utilize two sets of prompts at each layer for processing the spatial and high-frequency features respectively while freezing the parameters of the backbone. We denote the input features of the $i$-th layer as $\mathbf{F}_{i-1}^{RGB}$ and $\mathbf{F}_{i-1}^{HFQ}$. They are first reshaped to $\mathbb{R}^{(H_{i-1} \times W_{i-1}) \times C_{i-1}}$, then are joined by prompts $\mathbf{P}_{i-1}^{RGB}$ and $\mathbf{P}_{i-1}^{HFQ} \in \mathbb{R}^{n_p \times C_{i-1}}$ respectively. Therefore, the procedure of each layer (Eq. 3) is altered as:

$$\begin{aligned} \mathbf{F}_i^{RGB} &= \text{Layer}_i\left(\left[\mathbf{P}_{i-1}^{RGB}, \mathbf{F}_{i-1}^{RGB}\right]\right), \\ \mathbf{F}_i^{HFQ} &= \text{Layer}_i\left(\left[\mathbf{P}_{i-1}^{HFQ}, \mathbf{F}_{i-1}^{HFQ}\right]\right), \end{aligned} \tag{4}$$

where $[\cdot]$ represents for Concat operation.

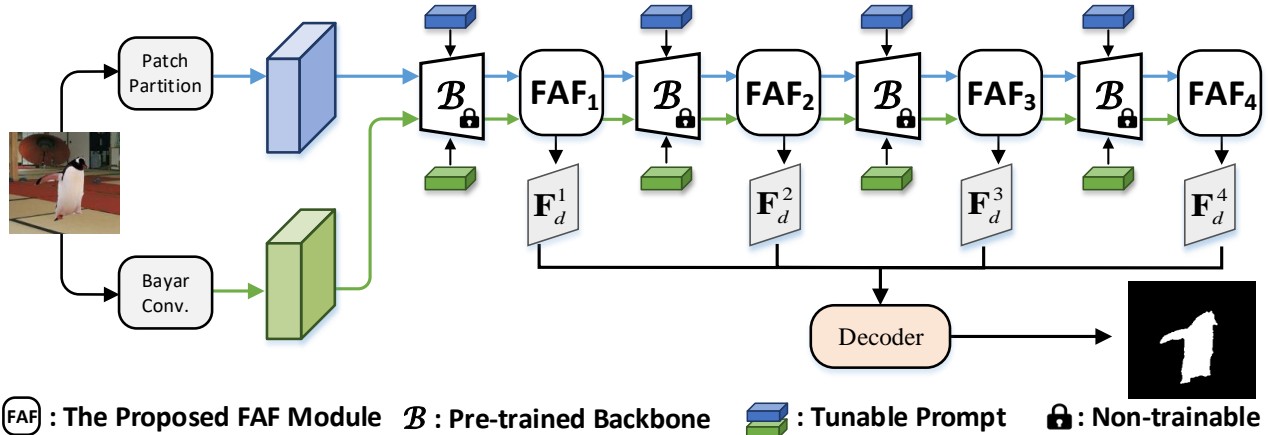

**Figure 2: Pipeline design of Prompt-IML. Single pre-trained backbone is frozen during training and tunable prompts are used to adjust features. The Feature Alignment and Fusion (FAF) module is a plug-and-play component for improving feature interaction and enhancement.**

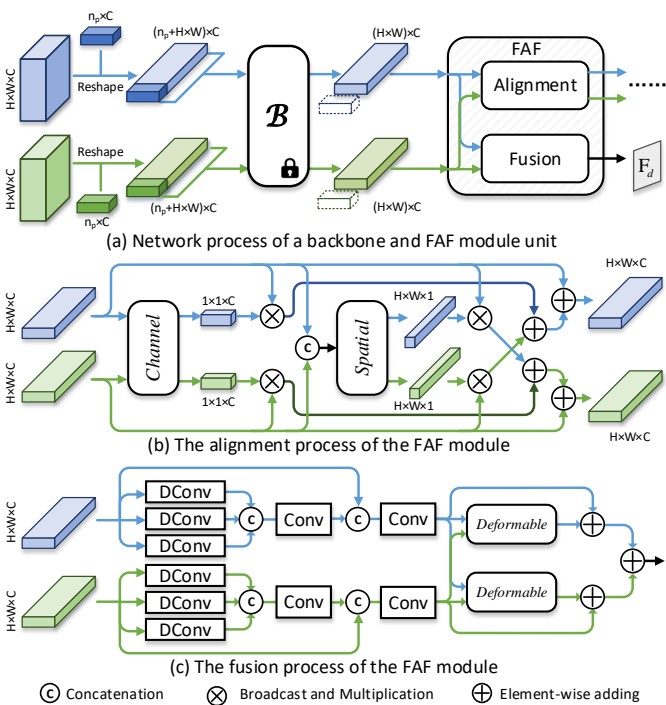

(a) Network process of a backbone and FAF module unit

(b) The alignment process of the FAF module

(c) The fusion process of the FAF module

ⓒ Concatenation    ⊗ Broadcast and Multiplication    ⊕ Element-wise adding

**Figure 3: Design of the proposed Feature Alignment and Fusion (FAF) module and usage of tunable prompts. The *Channel, Spatial, Deformable* represents the procedure of Eq. 5, Eq. 6, and Eq. 9.**

*3.2.2 Feature Alignment and Fusion Module.* Given spatial and high-frequency features processed by the backbone, we propose a FAF module for feature alignment and fusion. FAF modules are integrated between some adjacent layers of the backbone as shown in Fig. 2. The FAF module consists of an alignment stage[36] and a fusion stage, the detailed composition and procedure are depicted in Fig. 3.

In the feature alignment stage, we employ both channel attention and spatial attention to investigate the inter-channel and inter-spatial correlations of features, thereby enhancing features with corresponding information. Unprocessed features then gather information from enhanced features, reducing potential uncertainty and noise within itself. Specifically, we first employ average pooling operation (denoted by overline) to aggregate information. Then, they are concatenated on the dimension of $C_i$, which is denoted by $[\cdot]$, and fed into an MLP layer to generate channel-attention vectors $\mathbf{W}_i^{CRGB}, \mathbf{W}_i^{CHFQ} \in \mathbb{R}^{1 \times 1 \times C_i}$. The above procedure is formulated as:

$$
\begin{aligned}
\mathbf{W}_i^{CRGB}, \mathbf{W}_i^{CHFQ} &= \text{ChannelAttn}\left(\mathbf{F}_i^{RGB}, \mathbf{F}_i^{HFQ}\right) \\
&= \text{Split}\left(\text{MLP}\left(\left[\overline{\mathbf{F}_i^{RGB}}, \overline{\mathbf{F}_i^{HFQ}}\right]\right)\right),
\end{aligned}
\tag{5}
$$

where Split is the reverse operation of Concat. To obtain the spatial attention vector, we utilize two 1×1 convolutions with an intermediate ReLU layer, denoted by $g(\cdot)$, to aggregate spatial information. The procedure to obtain spatial-attention vectors $\mathbf{W}_i^{SRGB}, \mathbf{W}_i^{SHFQ} \in \mathbb{R}^{H_i \times W_i \times 1}$ can be formulated as:

$$
\begin{aligned}
\mathbf{W}_i^{SRGB}, \mathbf{W}_i^{SHFQ} &= \text{SpatialAttn}\left(\mathbf{F}_i^{RGB}, \mathbf{F}_i^{HFQ}\right) \\
&= \text{Split}\left(\text{Conv}\left(g\left(\text{Conv}\left(\left[\mathbf{F}_i^{RGB}, \mathbf{F}_i^{HFQ}\right]\right)\right)\right)\right).
\end{aligned}
\tag{6}
$$

Finally, we align the features from different branches by applying crosswise attention vectors, which produce the input for the next backbone layer through element-wise addition:

$$
\begin{aligned}
\mathbf{F}_i^{CRGB} &= \mathbf{W}_i^{CRGB} \odot \mathbf{F}_i^{RGB}, \quad \mathbf{F}_i^{SRGB} = \mathbf{W}_i^{SRGB} \odot \mathbf{F}_i^{RGB}, \\
\mathbf{F}_i^{CHFQ} &= \mathbf{W}_i^{CHFQ} \odot \mathbf{F}_i^{HFQ}, \quad \mathbf{F}_i^{SHFQ} = \mathbf{W}_i^{SHFQ} \odot \mathbf{F}_i^{HFQ}, \\
\mathbf{F}_i^{RGB} &:= \mathbf{F}_i^{RGB} + \mathbf{F}_i^{CHFQ} + \mathbf{F}_i^{SHFQ}, \\
\mathbf{F}_i^{HFQ} &:= \mathbf{F}_i^{HFQ} + \mathbf{F}_i^{CRGB} + \mathbf{F}_i^{SRGB}.
\end{aligned}
\tag{7}
$$

In the feature fusion stage, we first utilize dilated convolutions DConv with different dilation rates to process feature maps, enhancing interactions within patches. Specifically, we employ dilation

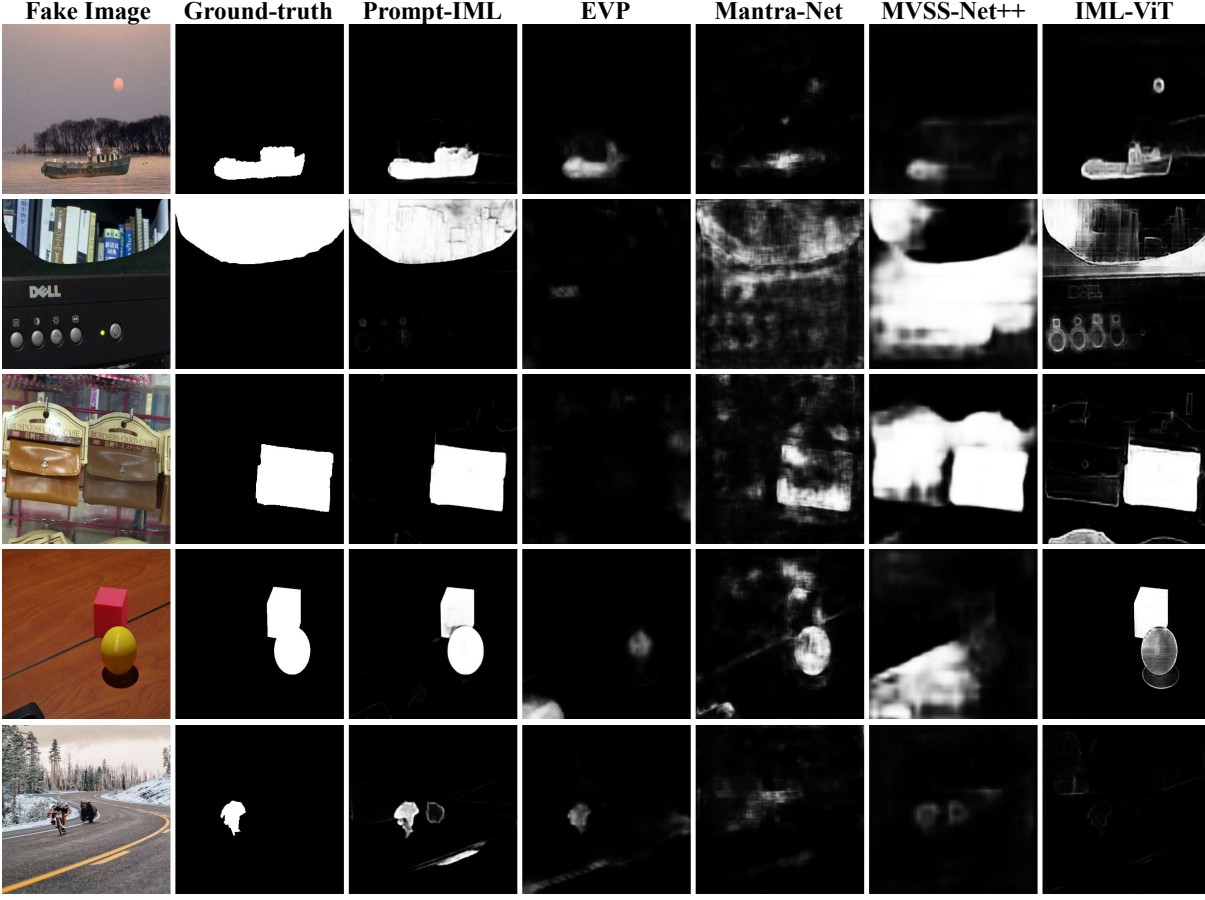

**Figure 4: Manipulation localization results on images originating from multiple datasets. Columns from left to right are: fake image, ground-truth, Prompt-IML, EVP, Mantra-Net, MVSS-Net++ and IML-ViT.**

rate $k \in \{1, 3, 5\}$ for processing, then concatenate outputs on the dimension of $C_i$. The concatenated features are processed to integrate information along with unprocessed features:

$$\tilde{\mathbf{F}}_i = \text{Conv}\left([\text{Conv}\left([\text{DConv}_{k \times k}\left(\mathbf{F}_i\right)]\right), \mathbf{F}_i]\right), k \in \{1, 3, 5\}. \quad (8)$$

Then, we apply deformable attention to facilitate the information interaction among patches from multi-views for fusion. The deformable attention mechanism not only reduces computational complexity through sampling with learnable offsets, but also helps suppress sporadic positive responses in feature maps, which aids localization since tampering operations typically affect specific regions of pixels rather than isolated ones[4]. Given the processed features from the previous step $\tilde{\mathbf{F}}_i^{RGB}$ and $\tilde{\mathbf{F}}_i^{HFQ}$:

$$\mathbf{attn}^{RGB} = \text{DeformAttn}_1\left(Q = \tilde{\mathbf{F}}_i^{RGB}, K\&V = \tilde{\mathbf{F}}_i^{HFQ}\right),$$

$$\mathbf{attn}^{HFQ} = \text{DeformAttn}_2\left(Q = \tilde{\mathbf{F}}_i^{HFQ}, K\&V = \tilde{\mathbf{F}}_i^{RGB}\right), \quad (9)$$

$$\mathbf{F}_i^d = \gamma_1 \cdot \left(\tilde{\mathbf{F}}_i^{RGB} + \mathbf{attn}^{RGB}\right) + \gamma_2 \cdot \left(\tilde{\mathbf{F}}_i^{HFQ} + \mathbf{attn}^{HFQ}\right),$$

where $\gamma_1, \gamma_2$ are learnable parameters. The output $\mathbf{F}_i^d$ is utilized for decoder.

### 3.3 Manipulation Localization Stage

To refine the multi-scale features obtained in the previous stage, we utilize the Mask2Former[3] as the decoder, which comprises two key components: a Pixel Decoder and a Transformer Decoder. The Pixel Decoder is responsible for progressively upsampling features from low to high resolution. The Transformer Decoder leverages query embeddings and multi-scale features for localization. This approach offers several advantages. Firstly, utilizing multi-scale features is advantageous for locating small tampered regions. Moreover, the integration of query embeddings with Masked-Attention helps constrain cross-attention solely to the tampered regions, thereby enhancing the extraction of tampering-related features.

### 3.4 Loss Function

Considering that the boundaries of tampered regions may exhibit jagged, non-smoothed edges and color inconsistencies, we draw inspiration from IML-ViT[22] and introduce edge supervision. Specifically, we use morphological operations, such as erosion and dilation, to process the mask $M$ and generate corresponding edge mask $M^\star$. In comparison to methods that utilize a network to generate edge predictions[4], this strategy not only incorporates edge information

**Table 1: Image Manipulation Localization Performance (F1 score with fixed threshold of 0.5). All methods are trained with CASIA2 for fair comparisons except for Mantra-Net and HP-FCN. We highlight the best and the second results in each column in bold and underlined respectively.**

| Method | Pixel-level F1 score | | | | | | |
|---|---|---|---|---|---|---|---|
| | CASIA1 | Columbia | NIST16 | COVER | DEF-12K | IMD20 | Average |
| HP-FCN, ICCV19[15] | 0.154 | 0.067 | 0.121 | 0.003 | 0.055 | 0.112 | 0.085 |
| Mantra-Net, CVPR19[33] | 0.155 | 0.364 | 0.000 | 0.286 | 0.155 | 0.187 | 0.191 |
| CR-CNN, ICME20[34] | 0.405 | 0.436 | 0.238 | 0.291 | 0.132 | 0.262 | 0.294 |
| GSR-Net, AAAI20[37] | 0.387 | 0.613 | 0.283 | 0.285 | 0.051 | 0.243 | 0.310 |
| MVSS-Net, ICCV21[2] | 0.452 | 0.638 | 0.292 | 0.453 | 0.137 | 0.260 | 0.372 |
| MVSS-Net++, PAMI22[4] | 0.513 | 0.660 | 0.304 | **0.482** | 0.095 | 0.270 | 0.387 |
| EVP, CVPR23[18] | 0.426 | 0.379 | 0.226 | 0.096 | 0.062 | 0.188 | 0.230 |
| IML-ViT, AAAI24[22] | 0.658 | 0.836 | 0.339 | 0.425 | 0.156 | 0.422 | 0.473 |
| Prompt-IML | **0.686** | **0.882** | **0.415** | 0.429 | **0.237** | **0.471** | **0.520** |

but also eliminates the need for adjustments to the backbone, enhancing its flexibility. The loss function comprises two components, each corresponding to the supervision of the prediction result and the edge:

$$\mathcal{L} = \mathcal{L}_{seg}(M_{gt}, M_{pred}) + \lambda \mathcal{L}_{edge}(M_{gt}^{\star}, M_{pred}^{\star}) \tag{10}$$

where $\mathcal{L}_{seg}$ and $\mathcal{L}_{edge}$ are binary cross-entropy functions, $M_{gt}$, $M_{gt}^{\star}$ denotes ground-truth mask and edge mask, and $M_{pred}$, $M_{pred}^{\star}$ symbolize predictions. $\lambda$ is a hyper-parameter and we set $\lambda = 20$ by default.

## 4 Experiments

### 4.1 Experimental Setup

*4.1.1 Datasets.* We adopt a common training protocol [2, 22, 37] of the IML task to facilitate fair comparisons of model performance and avoid the influence of private synthesis datasets. We solely utilize CASIA2 [5] to train Prompt-IML. 6 public test datasets, including CASIA1 [5], NIST16 [7], COVERAGE [32], Columbia [25], IMD2020 [26], and DEFACTO [23], are utilized for evaluation. Following MVSS-Net[2], we conduct testing on a sampled sub-dataset from DEFACTO, containing 6,000 genuine images and 6,000 manipulated images. The evaluation constitutes cross-dataset analysis, as there is no overlap between our training and test datasets.

*4.1.2 Evaluation Criteria.* We evaluate our model's performance on the test datasets using the pixel-level F1 score. Some previous methods employ the strategy of optimizing the F1 score with the optimal threshold, in which different thresholds are chosen for each image. However, the decision for optimal threshold necessitates ground-truth data, which is not feasible in real-world scenarios. Therefore, we report F1 score with fixed threshold, which is independent of the model itself and provides a fair assessment of model performance.

*4.1.3 Implementation Details.* We train our Prompt-IML on RTX 3090 GPUs for 80 epochs with a batch size of 2 in each GPU. Both the encoder and decoder are initialized with pre-trained weights on COCO[17]. Unless otherwise specified, all images are resized to $1024 \times 1024$. Following IML-ViT[22], we use simple and public

data augmentation techniques, including flipping, blurring, rotation, JPEG compression, randomly copy-moving, and inpainting rectangular areas within a single image. We use the AdamW[21] optimizer with a base learning rate of $1 \times 10^{-4}$ and schedule the learning rate utilizing a cosine decay strategy.

### 4.2 Performance Comparisons

We compare our method with the other 8 state-of-the-art methods to comprehensively evaluate our approach and report the F1 score in Tab. 1. We can observe that our method improves the best baselines with 2.8%, 4.6%, 7.6%, 8.1%, and 4.9% for each improved dataset, respectively. On average, it improves 4.3% compared to sub-optimal baseline IML-ViT [22]. These sufficiently demonstrate the superiority of our model. However, on COVER[32] dataset, MVSS-Net-based methods [2, 4] outperform all the other methods. COVER is a small-scale dataset of forged images created solely through copy-move techniques, with most detection clues located around the boundary of the forged regions. So, we attribute this phenomenon to their carefully designed edge information extraction structure and data augmentation techniques.

Furthermore, Fig. 4 showcases the predicted localization results of each model, with each image originating from a different dataset with substantial variations in the manipulated regions. The results underscore the remarkable generalization ability of our method, suggesting that the proposed approach can effectively leverage the prior knowledge embedded in pre-trained models to detect tampering traces.

### 4.3 Robustness

In this section, we utilize 6 test datasets to comprehensively evaluate the robustness of Prompt-IML. Following IML-ViT[22], we apply two common attack methods, i.e. JPEG compression and Gaussian blur, at various levels of perturbation, to create attacked images. The results are exhibited in Fig. 5.

In the JPEG compression test, the proposed Prompt-IML keeps clear advantages on 4 datasets. On COVER and NIST16, our method performs closely to the leading method. In the Gaussian blur test, Prompt-IML significantly outperforms other methods on all datasets.

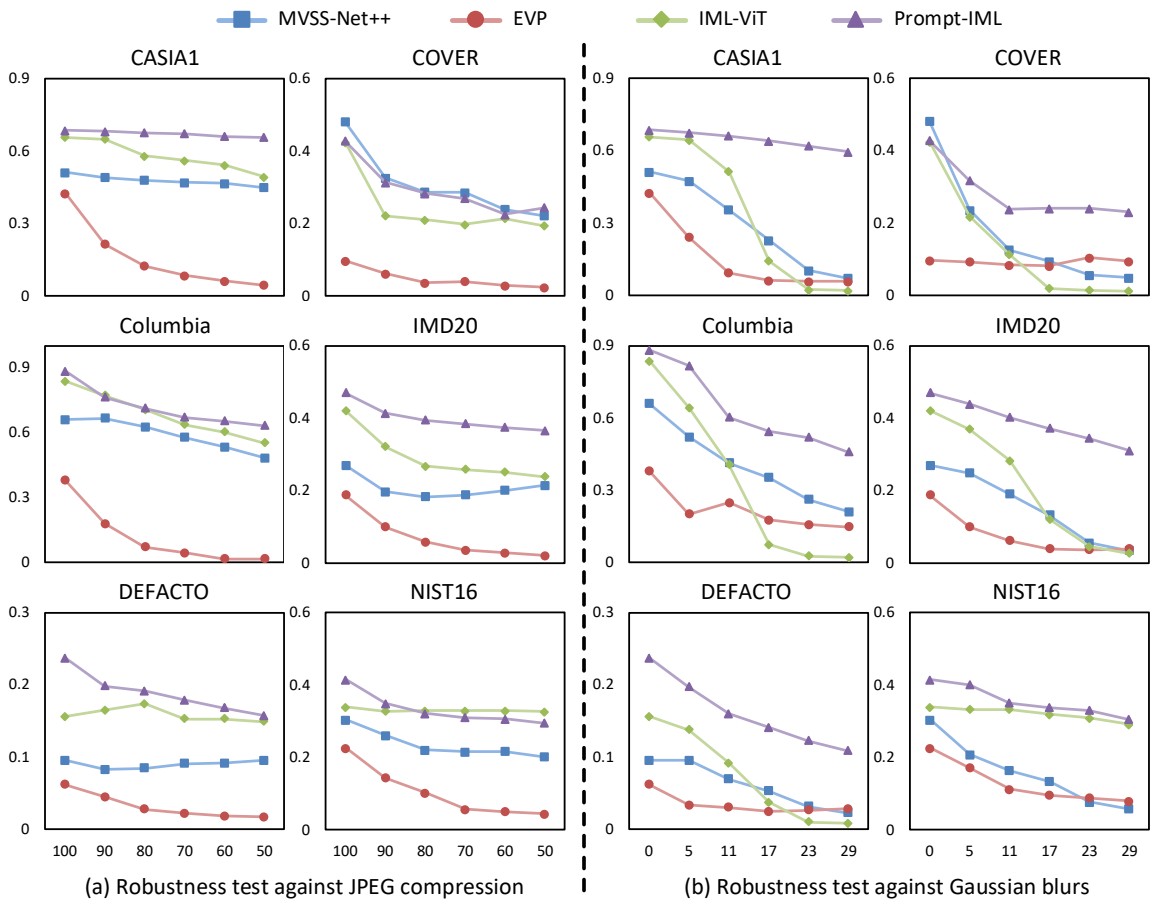

**Figure 5: Robustness evaluation against two image post-processing techniques, i.e. JPEG compression and Gaussian blurs. We report the processing intensity on the x-axis and the F1 score on the y-axis.**

**Table 2: Ablation study of Prompt-IML. We ablate spatial information (SP), high-frequency information (HF), alignment stage of FAF (ALN), and fusion stage of FAF (FSN) for study.**

| | components | | | | Pixel-level F1 score | | |
|---|---|---|---|---|---|---|---|
| | **SP** | **HF** | **ALN** | **FSN** | **COVER** | **NIST16** | **IMD20** |
| 1 | ✓ | - | - | - | 0.338 | 0.346 | 0.374 |
| 2 | ✓ | ✓ | - | - | 0.363 | 0.365 | 0.401 |
| 3 | ✓ | ✓ | ✓ | - | 0.399 | 0.384 | 0.402 |
| 4 | ✓ | ✓ | - | ✓ | 0.393 | 0.386 | 0.441 |
| 5 | ✓ | ✓ | ✓ | ✓ | **0.429** | **0.415** | **0.471** |

Overall, in comparison to other methods, Prompt-IML presents a remarkable ability to withstand both JPEG compression and Gaussian blur, especially against the latter. We also notice that IML-ViT exhibits better average robustness than other methods, so we attribute the robustness of our approach to the more effective utilization of large-scale pre-trained models, since these models can learn more robust features due to the extensive training datasets they are exposed to.

It's worth noting that our method exhibits a significant performance improvement compared to IML-ViT in resisting Gaussian blur attack. We believe these advantages stem from the use of high-frequency features and promot-tuning, leading to the following speculations. Firstly, IML-ViT fully fine-tunes the pre-trained network, which may harm its robustness due to catastrophic forgetting [30]. Additionally, the resistance of different features to various attacks vary, so leveraging multi-view features adequately may contributes to enhancing the robustness of the method.

## 4.4 Ablation Studies

We conduct several experiments following the settings outlined in Tab. 2, to thoroughly assess the effectiveness of the modules in our approach. We report the F1 score of each model on COVER [32], NIST16 [7], and IMD20 [26].

*4.4.1 Influence of Multi-view Features.* In setting 2, we employ a single backbone with frozen parameters to simultaneously extract both spatial and high-frequency features from images. Compared to setting 1, which only employs spatial features, we observe that the utilization of high-frequency features increases 2.5%, 1.9%, and 2.7%

**Table 3: Pre-trained backbone comparisons of Prompt-IML. All experiments are conducted using $512 \times 512$ manipulated images as input, and designed modules are slightly modified to match the requirements of feature map sizes.**

| Backbone | Pixel-level F1 score | | | | | |
|---|---|---|---|---|---|---|
| | CASIA1 | Columbia | COVER | NIST16 | IMD20 | DEF-12K |
| CLIP[28] | 0.522 | 0.528 | 0.185 | 0.269 | 0.241 | 0.077 |
| MAE[9] | 0.538 | 0.528 | 0.256 | 0.272 | 0.285 | 0.133 |
| SAM[13] | 0.596 | 0.481 | 0.228 | 0.281 | 0.219 | 0.094 |
| Swin-Transformer[20] | **0.631** | **0.814** | **0.278** | **0.350** | **0.332** | **0.177** |

**Table 4: The comparisons between Prompt Tuning and Full Tuning. In Full Tuning, we modify the model into a dual-branch structure to handle multi-view features and initialize all branches with pre-trained parameters.**

| Method | Pixel-level F1 score | | | | | |
|---|---|---|---|---|---|---|
| | CASIA1 | Columbia | COVER | NIST16 | IMD20 | DEF-12K |
| Prompt-Tuning | 0.686 | 0.882 | 0.429 | 0.415 | 0.471 | 0.237 |
| Full-Tuning | 0.702 | 0.885 | 0.610 | 0.414 | 0.542 | 0.280 |

in F1 scores respectively, effectively demonstrating the feasibility of exploiting pre-trained models to process multi-view features.

*4.4.2 Influence of FAF Module.* The proposed FAF module comprises two independent stages: alignment and fusion. So setting 3 and 4 are used to validate the effectiveness of each stage respectively. In setting 4, we skip the feature alignment stage and directly pass the features to the next layer. Compared with setting 5, we note a decrease in F1 scores on all three datasets when the feature alignment stage is absent, resulting in decreases of 3.6%, 2.9%, and 3.0% respectively. In setting 3, we skip the feature fusion stage and directly add multi-view features as fused features. Compared with setting 5, the absence of the feature fusion stage leads to decreases of 3.0%, 3.1%, and 7.1% in F1 scores individually. These results effectively demonstrate that the FAF module successfully enhances features through information interaction between features.

## 4.5 Choice of Pre-trianed Backbone

We investigate the impact of selecting different pre-trained models as the backbone. We utilize CLIP [28], MAE [9], SAM[13] and Swin-Transformer [20]. Both CLIP and MAE adopt architectures of the plain ViT, while SAM is similar to Swin-Transformer. Given the computational demands of the global self-attention mechanism especially for large images, we resize all images to $512 \times 512$ for this comparison. Moreover, due to the fixed feature map size output by the plain ViT, we incorporate several convolutions at the end of each Fusion stage within the FAF module to align with the input requirements of the decoder. We report the F1 scores for different backbones in Tab. 3. The Swin-Transformer model, which is trained on the COCO [17] dataset for semantic segmentation task, deliver superior results. We attribute this success primarily to its varied receptive fields across different layers, enabling it to uncover subtle tampering traces. Although CLIP is pre-trained on a large dataset, it emphasizes alignment between text and image features, hence using the image encoder alone may not be the optimal choice. Besides, we hypothesis that the implementation of the window attention mechanism in SAM may limit its performance on lower-resolution

images. Therefore, we select pre-trained Swin-Transformers as our backbone.

## 4.6 Prompt Tuning v.s. Full Tuning

We compare two approaches, prompt tuning and full tuning, for adapting pre-trained models to the IML task and assess their impact on model performance. When using the full tuning method, as a single backbone faces limitations in processing spatial and high-frequency features simultaneously, we adjust the backbone to a dual-branch architecture following the guidelines in [2, 14]. Tab.4 presents the F1 scores of different tuning methods. Full tuning does not lead to significant performance gains on most datasets but shows a substantial 19.1% improvement on the COVER dataset. We attribute this anomaly to the small scale of the COVER dataset and its use of a single manipulation technique. Although full tuning exhibits a degree of performance improvement, the dual-branch structure introduces significantly more trainable parameters compared to tunable prompts. For a more straightforward comparison, we do not calculate the learnable parameters of the FAF module and decoder, because they are included in both methods. The learnable parameters' size for the tunable prompts is 0.09M, and for the dual-branch backbones in full tuning is 93.14M. Therefore, prompt tuning is more advantageous in adapting to the development of large models and handling multi-view features.

## 5 Conclusion

In this paper, we explore the potential of utilizing existing pre-trained models to address the scarcity of public available datasets in the IML task. We propose Prompt-IML, which utilizes a single pre-trained network to extract multi-view features through tunable prompts. A specially designed Feature Alignment and Fusion (FAF) module is employed to integrate multi-view features, effectively reducing noise and uncertainty in features, and suppressing sporadic positive responses. Extensive experiments on 6 test datasets demonstrate outstanding performance, better generalization ability, and higher robustness of Prompt-IML.

## Acknowledgments

This work was supported by the National Natural Science Foundation of China under Grants U20B2051, 62072114, U20A20178, U22B2047.

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
