# OpenReview forum: "Multi-view Feature Extraction via Tunable Prompts is Enough for Image Manipulation Localization"
_acmmm.org/ACMMM/2024/Conference — MM2024 Poster_

### Official Review · Reviewer_7Dyz · 2024-05-20

**Rating:** 4
**Confidence:** 2

**Summary:**

This paper proposed a prompts image manipulation localization method (Prompt-IML). Instead of training the entire network, Prompt-IML fixed the backbone model weights, and use tunable prompts to adjust features. A feature alignment and fusion (FAF) module is proposed to fuse features from RGB and High-frequency views. Experimental results show its superior performance on 6 datasets.

**Strengths:**

1. Training the tunable prompts rather than the entire network reduces the training load.
2. FAF enables the feature alignment and fusion of two different views.
3. Experimental results show its superior performance over other baseline methods.

**Limitations:**

1. The authors claim FAF can "reduce noise and uncertainty in features through interactive processing", but it is not verified in the paper. Although we can see from Table 3 that FAF can improve the model performance, I am still curious about how they reduce the noise and uncertainty.
2. Adding prompts to the intermediate features with concatenation will change the dimension of original features, can this method be applied to other backbones that only accept fixed-size intermediate features?
3. In Table 5, the full-tuning setting obtains much better performance than prompt-tuning on CASIA1, COVER, IMD20, and DEF-12K. In real world applications, the testing performance of image manipulation localization algorithms is much more important than training load, since the model training only happens in offline stage. So why the authors still choose prompt-tunning in the proposed method?
4. Only F1 metric is used in evaluation, which may introduce bias. Please also include AUC metric as in other papers [16][20].

**Suitability:**

2

---

### Official Review · Reviewer_YPgH · 2024-05-23

**Rating:** 2
**Confidence:** 3

**Summary:**

This paper investigates the potential of using existing pretrained models to address the lack of publicly available datasets in  Image Manipulation Localization (IML) tasks. The authors introduce Prompt-IML, which employs a single pretrained network for extracting multi-view features via prompt tuning. They utilize a specially designed Feature Alignment and Fusion (FAF) module to integrate these multi-view features, effectively reducing noise and uncertainty while suppressing sporadic positive responses. Extensive experiments conducted on six test datasets showcase the outstanding performance, improved generalization ability, and higher robustness of Prompt-IML.

**Strengths:**

The lack of common training data sets in Image processing localization (IML) directly hinders the performance of IML implementation models. To address this challenge, this paper proposes a Prompt-IML framework that leverages the rich prior knowledge of pre-trained visual models by adopting tunable prompts. In addition, a plug-and-play feature alignment and fusion (FAF) module is developed to reduce noise and uncertainty in features through interactive processing. The proposed method is novel and technically correct. The effectiveness of the proposed method was demonstrated by visualized comparison with the images detected by the existing Prompt-IML, EVP, Mantra-Net, MVSS-NET ++ and IML-ViT methods. This paper evaluates the proposed method in 6 public test datasets, including CASIA1, NIST16, COVERAGE, Columbia,  IMD2020, and DEFACTO by F1. The proposed method improves the best baselines with 2.8%, 4.6%, 7.6%, 8.1%, and 4.9% for each improved dataset, respectively.

**Limitations:**

(1) Although methods exist to introduce prompt for tampering image detection using pre-trained models, the use of spatial features and high-frequency features as hints in this paper is novel. However, the proposed second module is not novel enough and has no particularity. Specifically, the interactive fusion method of channel attention and spatial attention is more common. In addition, the fusion approach is not innovative. Therefore, the whole article is not innovative enough.
(2)Some technical details are unclear, such as the introduction of the pre-trian model.
(3) Comparative methods in recent years should be added to fully demonstrate the effectiveness of the proposed method.
(4) The dimensions of the two fusion features in formula (4) are different, so fusion cannot be realized.

**Suitability:**

3

---

### Official Review · Reviewer_9v2b · 2024-05-26

**Rating:** 4
**Confidence:** 4

**Summary:**

This paper proposes an adapter that can adapt the backbone of commonly used visual pre-trained Transformers to the task of image manipulation localization. The "Prompt" in the title is implemented based on the visual prompt from the VPT paper[11]. Further, a modular Feature Alignment and Fusion (FAF) component integrates effortlessly into the backbone to achieve this functionality.

**Strengths:**

- The proposed method possesses performance advantages, effectively combining information from the high-frequency and RGB domains.
- Possessing a plug-and-play module allows for effective utilization of the current wealth of visual pre-trained Transformers.
- It can achieve SoTA performance with a comparable amount of data.
- The article has smooth and coherent writing.

**Limitations:**

### Some unclear details:
In this part, it does not fully represent the weakness of the article; it is mainly where I feel the description is not clear.
- I'm not quite clear whether $F^{RGB}$ and $F^{HFQ}$ are concatenated and then input into the Backbone block, or if they are input separately in two instances. The key is whether these two different modalities influence each other through global attention in the Backbone layer, or if they each compute their own information independently. This also affects the design of the position embedding.
### Weakness:
- As a method that does not require full-parameter training, one significant advantage should be computational complexity, but the authors have not provided the corresponding analysis. Please provide reasons for not giving the corresponding analysis.
- As a method primarily focused on the adapter, the paper only provides comparisons with three methods: CLIP, MAE, and Swin, which is quite limited and these are all relatively early methods. It is suggested to add comparisons with newer methods such as SAM[A] and Dinov2[B]. Drawing comparisons with adapters in other fields, for example, LoRA[C] has demonstrated performance advantages across eight different backbones, a necessary number is required to prove the advantages of the proposed method as a plug-and-play adapter.
- The ablation study for VPT is not very sufficient, and the performance of VPT is influenced by the choice between deep prompt and shallow prompt (as shown in Fig. 2 of the VPT[11] paper). The authors have chosen to apply the deep prompt strategy in this paper; further analysis is required to support the selection.

**Overall, this article has significant engineering value, if the above questions about the technical details are resolved, I would be glad to raise my rating.**

### Citation
[A] Kirillov, A., Mintun, E., Ravi, N., Mao, H., Rolland, C., Gustafson, L., ... & Girshick, R. (2023). Segment anything. In Proceedings of the IEEE/CVF International Conference on Computer Vision (pp. 4015-4026).

[B] Oquab, M., Darcet, T., Moutakanni, T., Vo, H., Szafraniec, M., Khalidov, V., ... & Bojanowski, P. (2023). Dinov2: Learning robust visual features without supervision. arXiv preprint arXiv:2304.07193.

[C]Hu, E. J., Shen, Y., Wallis, P., Allen-Zhu, Z., Li, Y., Wang, S., ... & Chen, W. (2021). Lora: Low-rank adaptation of large language models. arXiv preprint arXiv:2106.09685.

**Suitability:**

3

---

### Meta-Review · Area_Chair_od7j · 2024-07-03

**Recommendation:** Accept (Poster)
**Confidence:** 3

**Metareview:**

Two reviewers recommended weakly accept, and one recommended weakly reject and didn't make final rating. I recommend to accept the paper as a poster.